# Reverse-correlation analysis of navigation dynamics in *Drosophila* larva using optogenetics

Luis Hernandez-Nunez[1,2], Jonas Belina[3], Mason Klein[2], Guangwei Si[2], Lindsey Claus[2], John R Carlson[3], Aravinthan DT Samuel[2]*

[1]Center for Systems Biology, Harvard University, Cambridge, United States; [2]Department of Physics and Center for Brain Science, Harvard University, Cambridge, United States; [3]Department of Molecular, Cellular, and Developmental Biology, Yale University, New Haven, United States

**Abstract** Neural circuits for behavior transform sensory inputs into motor outputs in patterns with strategic value. Determining how neurons along a sensorimotor circuit contribute to this transformation is central to understanding behavior. To do this, a quantitative framework to describe behavioral dynamics is needed. In this study, we built a high-throughput optogenetic system for *Drosophila* larva to quantify the sensorimotor transformations underlying navigational behavior. We express CsChrimson, a red-shifted variant of channelrhodopsin, in specific chemosensory neurons and expose large numbers of freely moving animals to random optogenetic activation patterns. We quantify their behavioral responses and use reverse-correlation analysis to uncover the linear and static nonlinear components of navigation dynamics as functions of optogenetic activation patterns of specific sensory neurons. We find that linear–nonlinear models accurately predict navigational decision-making for different optogenetic activation waveforms. We use our method to establish the valence and dynamics of navigation driven by optogenetic activation of different combinations of bitter-sensing gustatory neurons. Our method captures the dynamics of optogenetically induced behavior in compact, quantitative transformations that can be used to characterize circuits for sensorimotor processing and their contribution to navigational decision making.

*For correspondence: samuel@physics.harvard.edu

## Introduction

To successfully navigate their environments, animals transform sensory inputs into motor outputs in patterns that strategically orient themselves towards improving conditions. The navigational strategies of insect larvae represent a long-standing paradigm for studying the mechanisms of animal orientation (*Loeb, 1918*; *Mast, 1938*). The small size and simple nervous system of the *Drosophila* larva, combined with its powerful genetic toolbox and recent advances in optical neurophysiology and anatomical reconstruction of circuit structure and connectivity, opens the possibility of understanding the neural encoding of animal navigation from sensory inputs to motor outputs without gaps (*Saalfeld et al., 2012*). To accomplish this, a quantitative framework to describe navigation decision-making is needed. Such a framework can then be used to dissect the function of the neurons and circuits in charge of processing sensory information.

*Drosophila* larva navigation involves the regulation of transitions between two basic motor states, runs during which the animal moves forward using rhythmic peristaltic waves and turns during which the larva sweeps its head back and forth until it selects the direction of a new run (*Luo et al., 2010*; *Gomez-Marin et al., 2011*; *Gomez-Marin and Louis, 2012*) (*Figure 1A*). Attractive and repulsive responses can be estimated by the tendency of the larva to aggregate near or avoid an environmental

**eLife digest** Living organisms can sense their surroundings and respond in appropriate ways. For example, animals will often move towards the smell of food or away from potential threats, such as predators. However, it is not fully understood how an animal's nervous system is setup to allow sensory information to control how the animal navigates its environment.

Optogenetics is a technique that allows neuroscientists to control the activities of individual nerve cells in freely moving animals, simply by shining light on to them. Here, Hernandez-Nunez et al. have used optogenetics in fruit fly larvae to activate nerve cells that normally respond to smells and tastes, while the larvae's movements were tracked. Fruit fly larvae were chosen because they have a simple, but well-studied, nervous system. These larvae also move in two distinct ways: 'runs', in which a larva moves forward; and 'turns', during which a larva sweeps its head back and forth until it selects the direction of a new run.

The data from these experiments were quantified using a specific type of statistical analysis called 'reverse correlation' and used to build mathematical models that predict navigational behavior. This analysis of the experiments allowed Hernandez-Nunez et al. to reveal how specific sensory nerve cells can contribute to pathways that control an animal's navigation—and an independent study by Gepner, Mihovilovic Skanata et al. revealed similar results.

The approach of using optogenetics in combination with quantitative analysis, as used in these two independent studies, is now opening the door to a more complete understanding of the connections between the activity of sensory nerve cells and perception and behavior.

stimulus (*Kreher et al., 2008*). Attractive and repulsive responses can also be observed in the movement patterns of individual larvae (*Louis et al., 2007*; *Luo et al., 2010*; *Gershow et al., 2012*). When the larva encounters improving conditions over time, it lowers the likelihood of ending each run with a turn, thereby lengthening runs in favorable directions. When the larva encounters improving conditions during each head sweep of a turn, it increases the likelihood of starting a new run, thereby starting more runs in favorable directions. Thus, subjecting the larva to an attractant tends to suppress transitions from runs to turns and stimulate transitions from turns to runs; subjecting the larva to a repellant has the opposite effects.

Much progress has been made in understanding the molecular and cellular organization of the chemosensory system of the *Drosophila* larva, but how specific chemosensory neurons relay information to guide navigational movements remains poorly understood. (*Kreher et al., 2005*; *Vosshall and Stocker, 2007*; *Kreher et al., 2008*; *Kwon et al., 2011*). One challenge of studying chemotaxis is that it is difficult to provide sensory input to behaving animals with the flexibility, receptor specificity, and precision needed to build computational models of chemosensory-guided navigation. The recent development of a red-shifted version of channelrhodopsin, CsChrimson, which is activated at wavelengths that are invisible to the larva's phototaxis system, now allows us to specifically manipulate the activity of neurons in behaving animals with reliability and reproducibility (*Klapoetke et al., 2014*).

Here, we sought a mathematical characterization of the navigation dynamics evoked by optogenetic activation of different sets of neurons. We focus on the navigation driven by chemosensory inputs. Although the organization of the chemosensory periphery is well-defined, the quantitative mapping from sensory activity to behavioral dynamics has not yet been determined. To do this, we engineered a high-throughput experimental setup capable of recording the run and turn movements of freely moving larvae subjected to defined optogenetic activation of selected chemosensory neurons. By measuring large numbers of animals responding to defined random patterns of optogenetic stimulation, we were able to collect enough data to use reverse-correlation analysis to connect optogenetic activation patterns of sensory neurons with motor patterns (*Ringach and Shapley, 2004*). We used this information to build linear–nonlinear (LN) models that accurately predict behavioral dynamics in response to diverse patterns of optogenetic activation of sensory neurons (*Geffen et al., 2009*).

We used our method to study how the optogenetic activation of olfactory receptor neurons (ORNs) and different sets of gustatory receptor neurons (GRNs) map to navigational movements. Analysis of gustatory neurons allowed us to investigate the navigational responses evoked by individual GRNs and

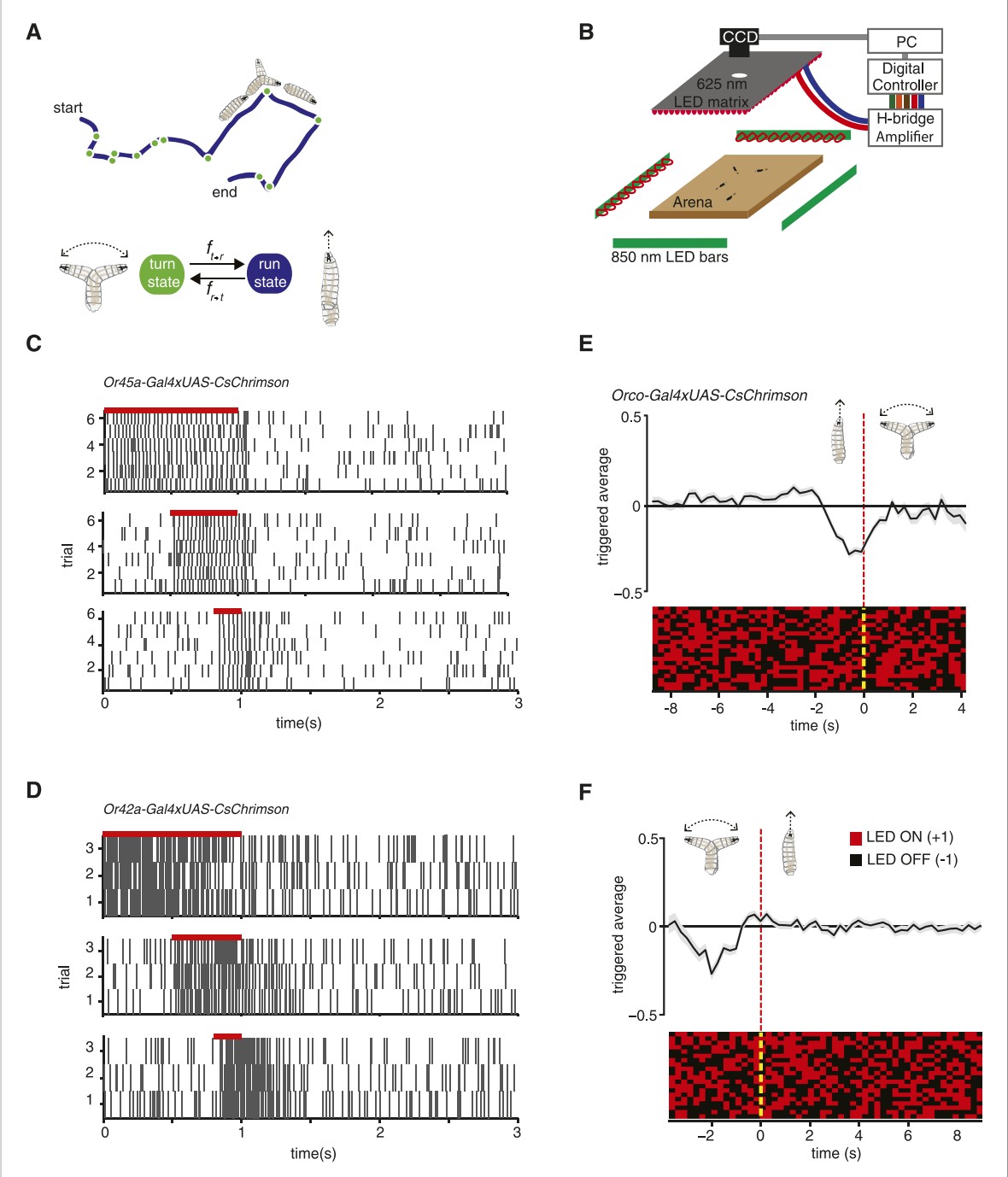

**Figure 1**. Experimental method for reverse-correlation analysis using optogenetics. (**A**) Larvae navigate by alternating between two basic motor states: runs and turns. The navigation strategy of the animal can be characterized by finding the mathematical functions, $f_{r \to t}$ and $f_{t \to r}$ that represent the stimulus dependence of transition rates. (**B**) Schematic of experimental setup. Larvae crawl on a 22 × 22 cm agar plate. Dark-field illumination is provided by lateral infrared LED bars, and animal movements are recorded with a CCD camera equipped with an infrared long-pass filter. Optogenetic illumination is provided by a matrix of red 625-nm LEDs from above. (**C**) We made extracellular recordings in the olfactory organ of the *Drosophila* larvae. Here, we show the rasters of the spikes induced by CsChrimson activation of the Or45a-expressing olfactory receptor neuron (ORN). We used 3 different pulse widths: 0.2, 0.5, and 1 s, all of them with the same intensity used for behavior experiments (1.9 W/m²). The red bar in the top of each raster represents the period during which red lights were ON. Each vertical line in the raster represents one spike. (**D**) Analogous to figure (**C**), we measured induced spiking of Or42a. The red bar in the top of each raster represents the period during which red lights were ON. Each vertical line in the raster represents one spike. (**E**) Mean stimulus history before each run-to-turn transition and (**F**) turn-to-run transition exhibited by *Orco>CsChrimson* larvae subjected to random ON/OFF optogenetic stimulation. The stimulus history for each motor state transition is aligned (dotted line) and averaged by assigning +1 to the LED

*Figure 1. continued on next page*

*Figure 1. Continued*

ON state and −1 to the LED OFF state. Data represent mean (black line) ± one SEM (gray shaded region) for 2018 transitions exhibited by 135 larvae. 20 event-triggered stimulus histories are shown in the raster to illustrate the random binary stimulus pattern used in our experiments.
The following figure supplement is available for figure 1:

**Figure supplement 1**. Optogenetic activation of OK6-Gal4 motor neurons.

their combinations. We find that compact LN models that connect optogenetic activation to behavioral responses are nonetheless sufficient to describe or predict navigational behavior and should facilitate future studies to elucidate the circuit mechanisms that shape sensorimotor transformations.

## Results

### Reverse-correlation analysis of navigation dynamics

We can characterize the navigation strategy of the *Drosophila* larva by identifying the mathematical functions that describe transitions between two basic motor states: running and turning (*Figure 1A*). We sought these functions ($f_{r \to t}$, $f_{t \to r}$) for defined patterns of chemosensory stimulation delivered via optogenetics. We used transgenic animals that express the red-shifted channelrhodopsin CsChrimson in selected olfactory and gustatory neurons using the UAS-Gal4 system (*Brand and Perrimon, 1993*). In our setup, we followed the movements of large numbers of late second-instar larvae navigating the surface of a 22 cm × 22 cm agar plate under dark-field illumination provided by infrared LEDs (*Figure 1B*). The entire plate was subjected to spatially uniform optogenetic illumination from above using a matrix of 625 nm red LEDs, a wavelength chosen to activate CsChrimson while invisible to the larva's photosensory system (*Keene and Sprecher, 2012*; *Klapoetke et al., 2014*). We tuned our light intensity (1.9 W/m$^2$) to a level where negligible behavioral response is detected in wild-type animals crossed with *UAS-CsChrimson* fed with 0.5 mM all-trans-retinal. We made sure that this light intensity is strong enough to activate CsChrimson by testing it with a well-studied motor neuron line (*Figure 1—figure supplement 1*).

To obtain direct evidence that optogenetic illumination in our behavioral setup activates sensory neurons, we used electrophysiology. We made extracellular recordings of the dorsal organ (DO) of individual larvae expressing CsChrimson in specific ORNs and recorded the responses to red light activation pulses of 0.2, 0.5, and 1 s of the same intensity used in the behavioral experiments. We found that optogenetic activation of the ORN-expressing Or45a reliably and reproducibly induced spike trains during exposure to red light (*Figure 1C*). Similar results were obtained using larvae expressing CsChrimson in the ORN-expressing Or42a (*Figure 1D*). These results confirm direct correspondence between ON/OFF pulses of CsChrimson activation and induced spiking in single sensory neurons.

To map the input–output relationships with optogenetic interrogation of chemosensory neurons, we used reverse-correlation methods viewing the whole animal as a transducer. We subjected larvae to random patterns of optogenetic stimulation and collected the statistics of all behavioral responses exhibited by the freely moving larvae. We used the simplest white process for reverse-correlation, a Bernoulli process where we assigned −1 for lights OFF and +1 for lights ON, and calculated the mean stimulus history that preceded each run-to-turn or turn-to-run transition (*Figure 1E,F*). These event-triggered stimulus histories represent how the animal uses optogenetic activation patterns of specific neurons to regulate each motor state transition and are proportional to the linear filter components of $f_{r \to t}$ and $f_{t \to r}$ (see 'Materials and methods').

### Linear filters of repellant and attractive responses

When freely crawling larvae encounter increasing chemoattractant concentrations during runs, they decrease the likelihood of initiating a turn. When they encounter increasing chemoattractant concentrations during the head sweep of a turn, they increase the likelihood of starting a new run. The Or42a receptor is activated by a number of volatile chemoattractants including ethyl butyrate and ethyl acetate (*Louis et al., 2007*; *Kreher et al., 2008*; *Asahina et al., 2009*). Genetically modified animals in which the *Or42a*-expressing ORN is the only functional ORN are capable of climbing olfactory gradients towards these attractants. These observations strongly suggest that the

*Or42a*-expressing ORN mediates attractant responses. To test our system, we subjected *Or42a>CsChrimson* larvae to random optogenetic stimulation. We found that run-to-turn transitions coincided with a decrease in the probability of optogenetic activation (**Figure 2A**, left), whereas

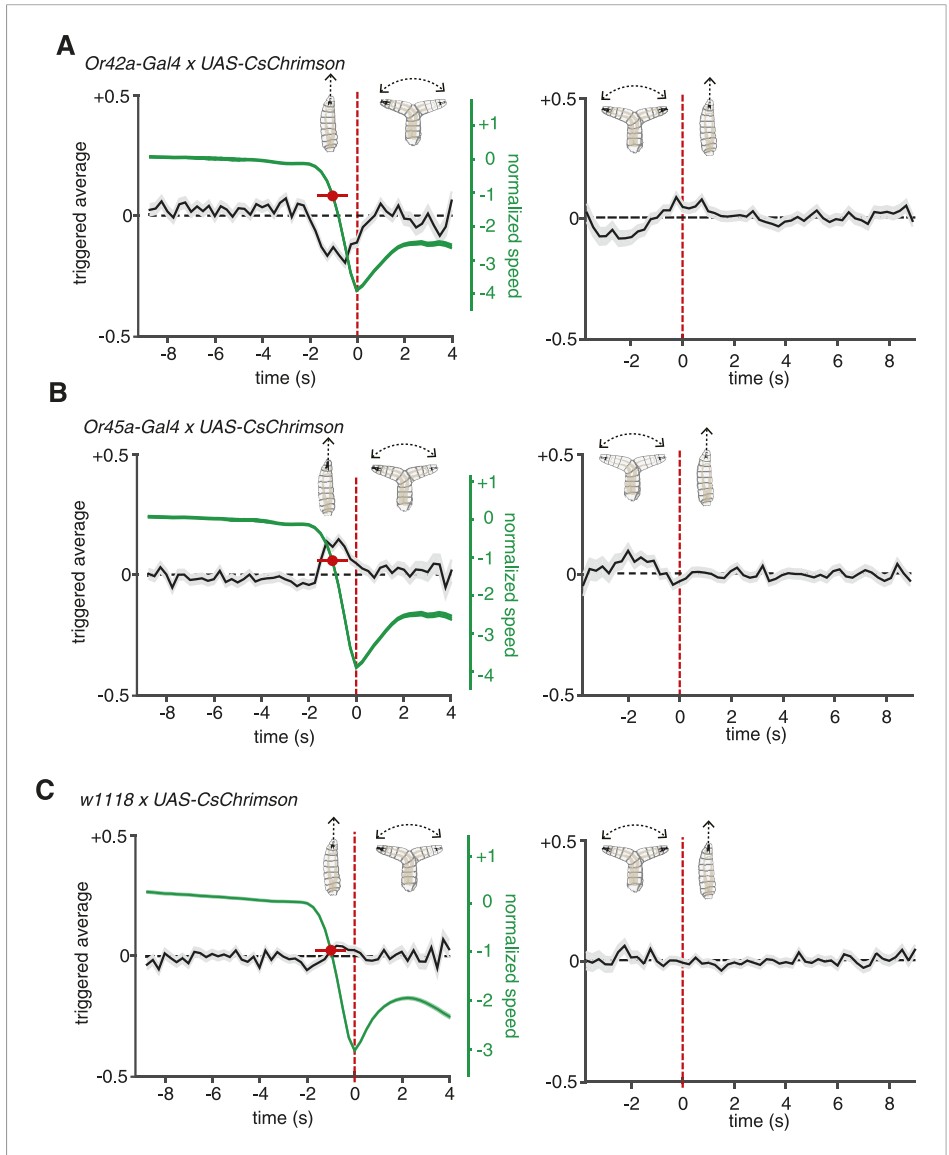

**Figure 2**. ORNs evoked navigation strategy. (**A**) Event-triggered stimulus histories for run-to-turn (left panel) and turn-to-run (right panel) transitions exhibited by *Or42a>CsChrimson* larvae subjected to random optogenetic stimulation as described in **Figure 1**. Consistent with an attractive response, the likelihood of optogenetic activation falls before a run-to-turn transition and rises before a turn-to-run transition. In run-to-turn transitions, crawling speed begins to fall before the initiation of turning movements (green traces). The mean beginning of deceleration averaged over all animals is flagged by the red dot (± STD). The units of normalized speed are standard deviations away from the mean crawling speed during runs. Data represent mean (black line) ± one SEM (grey shaded region) for 2752 transitions exhibited by 124 larvae. (**B**) Event-triggered stimulus histories exhibited by *Or45a>CsChrimson* larvae. Consistent with a repulsive response, the likelihood of optogenetic activation increases before a run-to-turn transition and decreases before a turn-to-run transition. Data represent mean (black line) ± one SEM (gray shaded region) for 3313 transitions exhibited by 119 larvae. The mean beginning of deceleration averaged over all animals is flagged by the red dot (± STD). (**C**) Control larvae event-triggered averages. Event-triggered averages of control larvae were uncorrelated with red light illumination patterns. Data represent mean (black line) ± one SEM (gray shaded region) for 4677 transitions exhibited by 121 larvae. The mean beginning of deceleration averaged over all animals is flagged by the red dot (± STD).

turn-to-run transitions coincided with an increase (*Figure 2A*, right). These patterns are consistent with an attractive response to Or42a activation. Importantly, the full shape of the event-triggered stimulus histories informs us about how the temporal optogenetic activation patterns of Or42a regulate each type of navigational movement. Methods that measure the tendency of larvae to aggregate near chemoattractants or net movement up chemoattractant gradients provide information about the overall tendency to navigate but not about the discrete decision-making processes that drive navigation.

Random optogenetic activation of all the ORNs via expression of *UAS-CsChrimson* with the *Orco* olfactory-receptor-coreceptor driver (previously called *Or83b*) mediated an attractive response similar to the one shown with the *Or42a* driver alone (*Figure 1E,F*). This is consistent with most ORNs in the *Drosophila* larva being thought to mediate attractant responses (*Kreher et al., 2008*; *Mathew et al., 2013*). One exception is the *Or45a*-expressing ORN, which has recently been shown to mediate an aversive response in an optogenetic setup; larvae that express channelrhodopsin in *Or45a*-expressing neurons will avoid an illuminated region of an agar plate (*Bellmann et al., 2010*). A role for the *Or45a*-expressing neurons in repellency is also consistent with the observation that they are the only ORNs that detect octyl acetate, a chemical repellant (*Cobb and Dannet, 1994*; *Kreher et al., 2008*). We sought the linear filters of this olfactory repellant response in our setup by quantifying the movements of *Or45a>CsChrimson* larvae subjected to random optogenetic stimulation (*Figure 2B*). Run-to-turn transitions in *Or45a>CsChrimson* larvae coincided with an increase in the probability of optogenetic illumination and turn-to-run transitions coincided with a decrease, consistent with repellant behavior.

For comparison, we calculated event-triggered stimulus histories using larvae heterozygous for UAS-CsChrimson and with the same genetic background as our Gal4 lines (w1118 × UAS-*CsChrimson*) subjected to random illumination (*Figure 2C*). These control larvae were raised in the same conditions and fed the same food as larvae used for all other experiments ('Materials and methods'). These larvae showed no correlations between the probability of illumination and motor state transitions. Their motor state transitions were random and spontaneous.

In our setup, we flag turn-to-run transitions by the resumption of peristaltic forward movement and run-to-turn transitions as the onset of head-sweeping behavior. However, the decision to finish a run may begin at an earlier point, when the animal first begins to slow down. We measured the crawling speeds of larvae before flagged run-to-turn transitions and found that runs decelerate ~1 s before the onset of head-sweeping behavior (*Figure 2A–C*). For both repellant and attractants, an increase and decrease in the probability of optogenetic illumination, respectively, coincides with the beginning of run deceleration (*Figure 2A,B*). The average deceleration time was 1 s for all experiments conducted in this study (Student's t-test, p < 0.01).

## LN models of behavior

A satisfactory model of navigation should be able to predict behavioral responses to various stimulus waveforms. We asked whether we could use our measurements of event-triggered stimulus histories to build such a model. A simple and widely used formalism is the LN model. In LN models, the linear filter component is proportional to our measurement of the event-triggered stimulus history (*Geffen et al., 2009*). First, the stimulus waveform is passed through this linear filter to make an initial prediction of the behavioral response. Linear estimates have common problems, such as taking negative values and failing to account for saturation. To correct these problems, the linear prediction is then scaled with a static nonlinear function. This static nonlinearity can be calculated by comparing a linear prediction with experimental measurements.

We used larvae with CsChrimson in *Or45a*-expressing neurons to test an LN model in predicting behavior. We calculated the static nonlinearity for both run-to-turn and turn-to-run transitions by comparing linear predictions obtained with the event-triggered stimulus histories shown in *Figure 2B* with experimental measurements (*Figure 3A*). Next, we implemented the linear filter and static nonlinearity in an LN model (*Figure 3B*) to predict the behavioral response of these larvae to different inputs, using step increases in optogenetic illumination as well as defined trains of pulses of different widths. We found remarkably good agreement in these predictions to both stimulus types (*Figure 3C*). We note that the LN prediction begins to fail to account for the turn-to-run transitions at long times following a step increase in optogenetic illumination. Turns typically last <4 s, which limits the length of stimulus history that can be used in a linear filter, which thus puts a ~4-s upper bound on the length of stimulus response that can be predicted. Taken together, our results show that

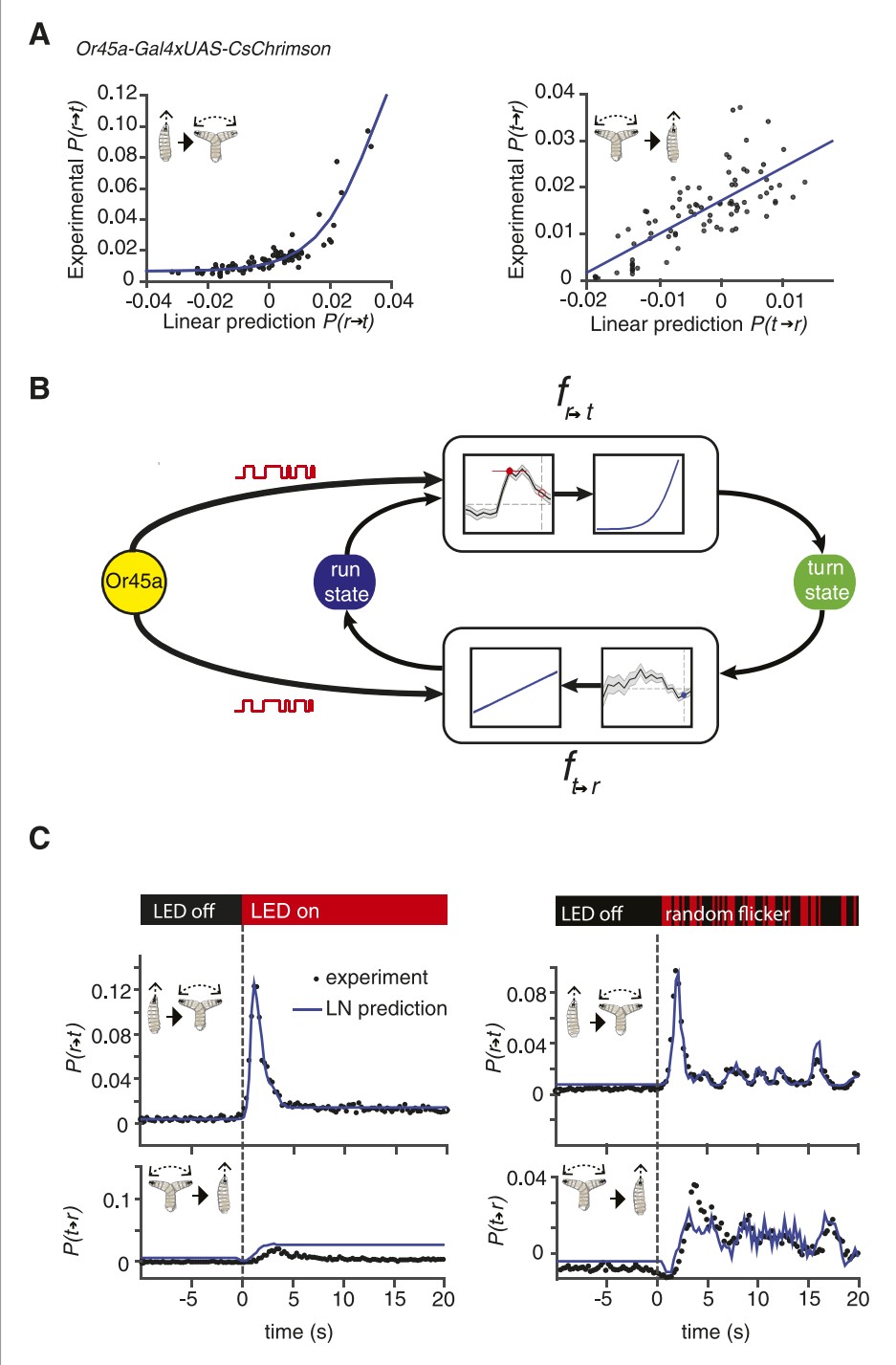

**Figure 3.** Linear-nonlinear (LN) models of behavior. (**A**) Estimating the static nonlinear function for run-to-turn and turn-to-run transitions exhibited by *Or45a>CsChrimson* larvae. Linear prediction using the event-triggered stimulus histories from *Figure 2B* was compared with the experimental measurements that generated the stimulus histories. The static nonlinearity for the run-to-turn transition is fitted using least squares estimation of a sigmoidal function ($R^2 = 0.8792$). The static nonlinearity for the turn-to-run transition is fitted with a line ($R^2 = 0.5041$). (**B**) Schematic representation of the LN model of navigation. Linear filters are convolved with the input signal, and the result is scaled according to the static nonlinear function fitted to estimate the probability rates for switching from one motor state to the other. (See 'Materials and methods'). (**C**) LN model predictions (blue lines) of behavioral responses to step changes in optogenetic illumination (left panels) and defined random flicker (right panels). Predictions are

*Figure 3. Continued*

made using the linear filter measured in *Figure 2B* and the static nonlinearity measured in *Figure 3A*. Experimental measurements to compare with prediction (black dots) represent data from N = 120 for the step response prediction and N = 240 larvae for the flicker response prediction.

The following figure supplements are available for figure 3:

**Figure supplement 1**. LN models of Gr21a and Gr10a.

**Figure supplement 2**. LN models of Or42a and Gr2a.

LN models governing stimulus-evoked transitions between motor states can be used to predict larval chemotaxis behavior with high accuracy (*Figure 3C*). The LN model was also successful in predicting the behavior of *Or42a*-expressing neurons and other chemosensory neurons (*Figure 3—figure supplement 1*; *Figure 3—figure supplement 2*; the procedure for detailed calculations are described in 'Materials and methods').

## Distinct temporal dynamics in optogenetically induced chemotactic behavior

The dynamics of behavioral responses are shaped by the linear filter component of LN models, while the static nonlinearity only provides saturation and instantaneous scaling. To test if optogenetic activation of different chemosensory neurons could produce behavioral responses with distinct dynamics, we undertook a search for different linear filters measured by event-triggered stimulus histories using reverse-correlation.

The Gr21a receptor senses carbon dioxide, a powerful *Drosophila* repellant (*Faucher et al., 2006*; *Gershow et al., 2012*). We measured the event-triggered stimulus histories of *Gr21a>CsChrimson* larvae subjected to random optogenetic stimulation. We found that run-to-turn transitions coincided with an increase in the probability of optogenetic illumination from baseline, whereas turn-to-run transitions coincided with a decrease (*Figure 4A*). These patterns are consistent with a repellant response. However, the linear filter associated with Gr21a for run-to-turn transition revealed important differences in shape and timing of stimulus history as compared with the filter for Or45a. The run-to-turn transition in both cases was preceded by a positive lobe in the probability of optogenetic activation lasting ~2 s. This positive lobe was itself preceded by a pronounced negative lobe lasting ~1.5 s for Gr21a but not for Or45a.

How do differences in the shape and timing of linear filters translate into behavioral responses with different dynamics? To explore this question, we compared the prediction and experimental measurement of stepwise activation of *Or45a*- and *Gr21a*-expressing neurons (*Figure 4B*). Biphasic linear filters—such as that associated with Gr21a and also seen in other sensory systems like the *Escherichia coli* chemotactic response—contribute to adaptation following transient stimulation (*Block et al., 1982*). A step increase in stimulation with repellants will cause a transient increase in the probability of run-to-turn transition. We predicted and confirmed differences in the adaptive return to baseline behavior for Gr21a and Or45a. The probability of run-to-turn transition returns to baseline faster in the case of Gr21a. Since each point represents a distribution of binary values (larvae transitioning from running to turning and larvae not transitioning), we used a z-test to identify regions where $P(r \rightarrow t)$ is significantly higher than baseline with p < 0.05. We found that $P(r \rightarrow t)$ of Gr21a larvae reach values significantly higher than baseline at least 0.5 s earlier than Or45a larvae. In addition, Or45a larvae stay at elevated values of $P(r \rightarrow t)$ for at least 0.75 s longer than Gr21a larvae (*Figure 4—figure supplement 1A*).

We also asked whether differences in behavioral dynamics caused by different linear filters might be found in attractant responses. Gr2a is expressed in the A1 and A2 GRNs of the DO as well as in two unidentified neurons in the terminal organ (*Kwon et al., 2011*). The role of the Gr2a receptor is not known, but it is part of the subfamily of Gr68a, which has been identified as a pheromone receptor in the adult fly (*Bray and Amrein, 2003*). We calculated the event-triggered stimulus histories of *Gr2a>CsChrimson* larvae and found that run-to-turn transitions coincided with a decrease in optogenetic activation, consistent with an attractant response (*Figure 4C*). Interestingly, the linear

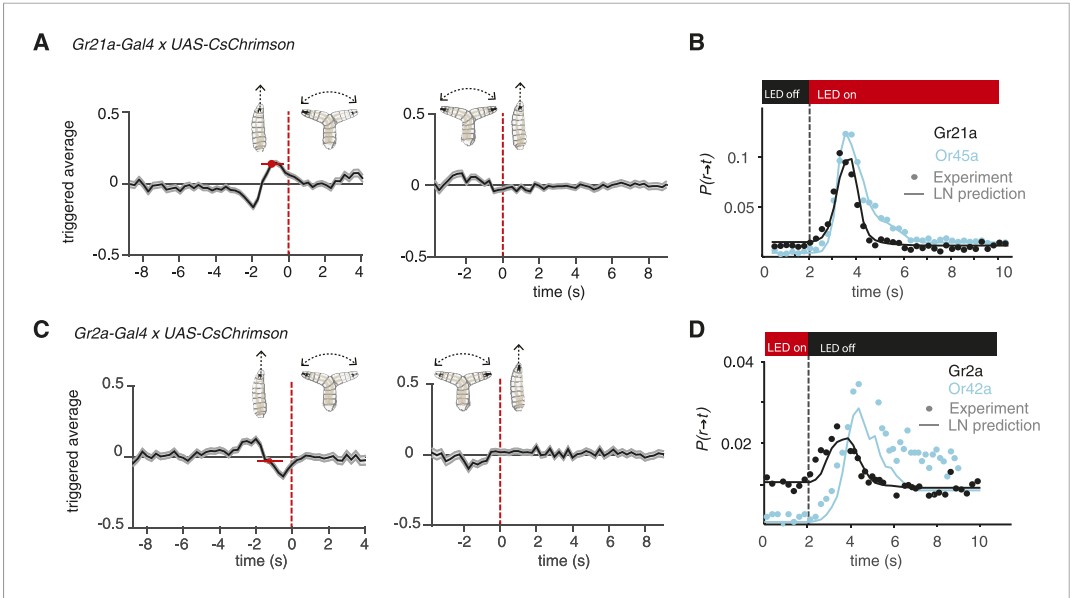

**Figure 4**. Distinct navigation dynamics. (**A**) Event-triggered stimulus histories exhibited by *Gr21a>CsChrimson* larvae. Linear filters of Gr21a neurons. Consistent with a repulsive response, the likelihood of optogenetic activation increases before a run-to-turn transition and decreases before a turn-to-run transition. Data represent mean (black line) ± one SEM (gray shaded region) for 4680 transitions exhibited by 90 larvae. The mean beginning of deceleration averaged over all animals is flagged by the red dot (± STD). (**B**) LN prediction and experimental measurements of different repellant responses to step changes in optogenetic illumination. Faster adaptation to baseline is observed in the case of the *Gr21a*-expressing neurons than *Or45a*. Step responses were measured with 115 Gr21>CsChrimson larvae and 120 Or45a>CsChrimson larvae; each larva was subjected to 30 steps of optogenetic activation. (z-test substantiate significant difference in the dynamics of the cyan and black curves see *Figure 4—figure supplement 1A*). (**C**) Event-triggered stimulus histories exhibited by *Gr2a>CsChrimson* larvae. Consistent with an attractive response, the likelihood of optogenetic activation decays before a run-to-turn transition and raises before a turn-to-run transition. Data represent mean (black line) ± one SEM (gray shaded region) for 3672 transitions exhibited by 128 larvae. The mean beginning of deceleration averaged over all animals is flagged by the red dot (± STD). (**D**) Linear prediction and experimental measurements of different attractant responses to step changes in optogenetic illumination. Faster responses and adaptation to baseline are observed in the case of the *Gr2a* than *Or42a*. Step responses were measured with 195 Gr2a>CsChrimson larvae and 117 Or42a>CsChrimson larvae; each larva was subjected to 30 steps of optogenetic activation. (z-test substantiate significant difference in the dynamics of the cyan and black curves see *Figure 4—figure supplement 1B*).

The following figure supplement is available for figure 4:

**Figure supplement 1**. Statistical analysis of behavioral dynamics.

filter associated with Gr2a was distinct from that of Or42a. In *Or42a>CsChrimson*, the run-to-turn transition was preceded by a single negative lobe lasting ~2 s. In *Gr2a>CsChrimson* larvae, the negative lobe was itself preceded by a positive lobe. As we did for repellants (*Figure 4B*), we asked whether the response dynamics to step decrease in optogenetic stimulation were distinct. We predicted and confirmed differences in behavioral dynamics. The most noticeable feature is that Or42a larvae reach different steady states of $P(r \rightarrow t)$ for lights ON or OFF; this creates differences in step-response dynamics. We conducted a z-test to identify regions where $P(r \rightarrow t)$ is significantly higher than baseline with $p < 0.05$. Since the steady-state $P(r \rightarrow t)$ for Or42a larvae is different for lights ON and lights OFF, we conducted the z-test with both values (*Figure 4—figure supplement 1B*). Because Or42a larvae start at a lower $P(r \rightarrow t)$, they take at least 1.75 s longer than Gr2a larvae for $P(r \rightarrow t)$ to become significantly higher than the lights OFF steady-state $P(r \rightarrow t)$. Comparison with the steady-state $P(r \rightarrow t)$ for lights ON confirms that the steady-state $P(r \rightarrow t)$ for lights OFF is significantly higher with $p < 0.05$ (*Figure 4—figure supplement 1B*).

We note that unlike the linear filters for run-to-turn transitions, the linear filters for turn-to-run transitions showed a similar shape for all Gal4 drivers that we used in this study. These filters only showed some variation in amplitude (*Figures 1, 2, 4, 5*).

## Navigational responses from bitter-sensing GRNs

The molecular and cellular organization of the chemosensory system of the *Drosophila* larva is numerically simple. The 21 ORNs contained in the larval DO together express 25 members of the Or family of odorant receptors and the Orco coreceptor (*Fishilevich et al., 2005*; *Kreher et al., 2005*). In contrast, 10 GRNs distributed in the DO and terminal organ—named A1, A2, B1, B2, and C1-C6—together express 28 members of the Gr family of gustatory receptors. Whereas most ORNs express a single Or, GRNs can express multiple Grs and each Gr can be expressed in multiple GRNs (*Kwon et al., 2011*). Thus, using larvae expressing CsChrimson under the control of different *Grx*-Gal4 drivers enabled us to assess the contribution of selected GRNs to behavior.

The C1 neuron expresses 17 receptors, some of which are found in other neurons (e.g., Gr32a, which is also found in B2) and some of which are specific to C1 (e.g., Gr9a). Most Grs are thought to respond to repulsive bitter compounds because they express the bitter markers Gr33a and Gr66a (*Kwon et al., 2011*), suggesting that C1 is a broadly tuned mediator of repellant responses. Consistent with this hypothesis, optogenetic activation of C1 with random stimuli using *Gr9a>CsChrimson* larvae evoked a weak repellant response where the run-to-turn transition coincided with a slight increase in the probability of optogenetic illumination (*Figure 5A*) (this

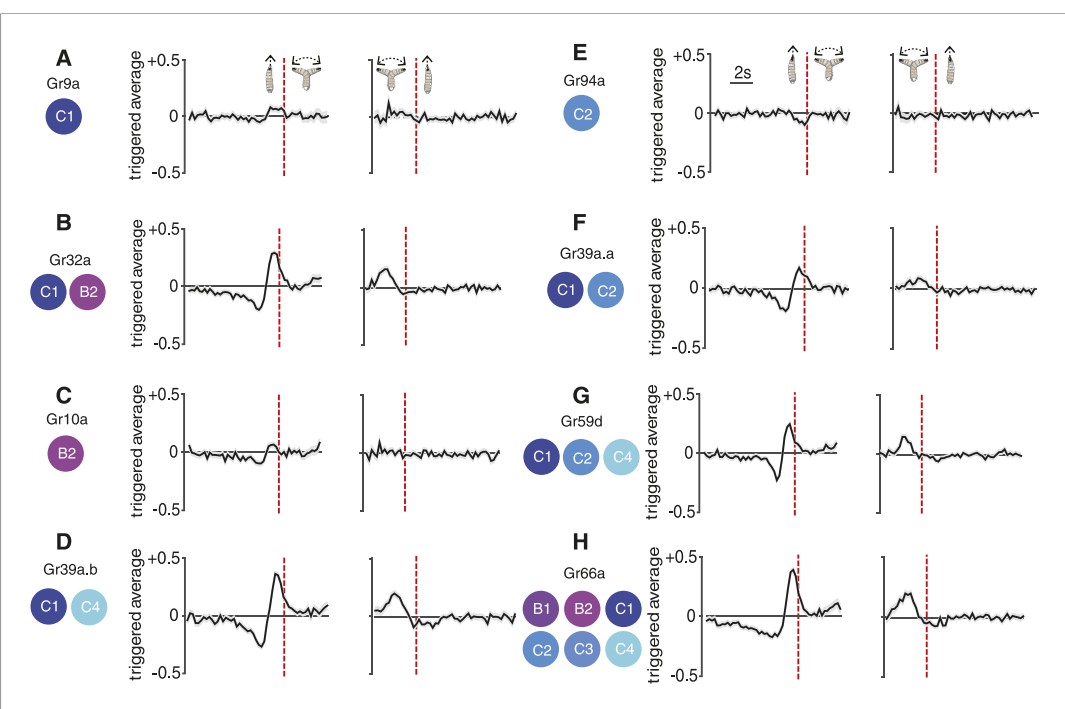

**Figure 5**. Reverse-correlation analysis of bitter-sensing gustatory receptor neurons (GRNs). Event-triggered stimulus histories exhibited by *GrX>CsChrimson* larvae using a set of GAL4 drivers that express in different subsets of GRNs. The cellular identities describing each expression pattern are taken from *Kwon et al. (2011)*. Each measurement represents 3270 to 4016 transitions taken from 87 to 134 larvae. Curves represent mean (black line) ± one SEM (gray shaded region).

The following figure supplements are available for figure 5:

**Figure supplement 1**. Statistical analysis of Gr9a and Gr94a triggered average.

**Figure supplement 2**. Normalized speed of Gr lines.

response was significantly different than the control with p < 0.05, see *Figure 5—figure supplement 1A*). The crawling speeds of larvae before flagged run-to-turn transitions triggered by optogenetic activation of GRNs is shown in *Figure 5—figure supplement 2*. Optogenetic activation of C1 together with B2 using *Gr32a>CsChrimson* larvae evoked a much stronger repellant response (*Figure 5B*). Optogenetic activation of specifically the B2 neuron using *Gr10>CsChrimson* larvae evoked a repellant response (*Figure 5C*). Optogenetic activation of C1 together with C4 using *Gr39a.b>CsChrimson* larvae generated a strong repellant response (*Figure 5D*). One possibility is that co-activation of narrowly tuned GRNs that express fewer Grs potentiates the repellant response of the broadly tuned C1 GRN; however, this interpretation should be taken with caution since different Gal4 drivers may induce different spiking rates upon optogenetic activation with CsChrimson.

We found that optogenetic activation of the C2 neuron alone using *Gr94a>CsChrimson* larvae evoked a weak attractive response (*Figure 5E*) (this response was significantly different than the control with p < 0.05, see *Figure 5—figure supplement 1B*). This is surprising because the C2 neuron also expresses the bitter receptors Gr33a and Gr66a, which should drive repellant responses, although these receptors are also found in other neurons. One possibility is that the attractant response driven by C2 is inverted when additional gustatory neurons are recruited. This hypothesis is supported by our observation that co-activation of C1 and C2 using *Gr39a.a>CsChrimson* larvae exhibited a much stronger repellant response than activation of C1 alone (*Figure 5F*). Co-activation of C1, C2, and C4 using *Gr59d>CsChrimson* larvae also exhibited a strong repellant response (*Figure 5G*). The strongest repellant response was observed by co-activating C1-C4, B1, and B2 using *Gr66a>CsChrimson* larvae (*Figure 5H*).

## Discussion

A fundamental step towards understanding how animal navigation is encoded in neural circuits is the development of a quantitative framework that accurately describes behavioral dynamics. To take this step with the *Drosophila* larva, we combined optogenetics with high-resolution behavioral analysis and reverse-correlation techniques to build LN models that provide an accurate estimate of the decision-making processes that guide navigation during optogenetically induced chemotaxis.

LN models separate time dependencies and instantaneous scaling into two modules, the linear filter and static nonlinearity, respectively. We find that the LN model is capable of accounting for diverse dynamics across attractant and repellant responses in both the gustatory and olfactory systems. For example, LN models accurately predicted the differences in response speed and adaptation when different GRNs and ORNs were activated. One reason for the diversity of dynamics is that the *Drosophila* larva chemosensory system, in addition to encoding attractant and repellant responses, is also capable of shaping the dynamics of behavioral responses in ecologically important ways. For example, the priorities given to specific chemicals encountered in the environment might not only be measured in terms of their relative degrees of attraction or repulsion but also in the speed of the behavioral response that they trigger or the speed of adaptation. We note that some of the observed differences in behavioral dynamics might be caused by using different transgenic lines and different Gal4 drivers with different potencies. It would thus be useful to confirm the differences in behavioral dynamics that are suggested by our optogenetic manipulations with direct stimulation of each GRN and ORN and quantitative behavioral analysis in defined environments using cell-specific odorants and tastants.

Navigational dynamics evoked by specific sets of gustatory neurons have remained elusive because of the lack of chemicals that are specific to individual GRNs. Our reverse-correlation analysis using optogenetic activation with CsChrimson allowed us to determine not only the valence (attraction or repulsion) of navigation mediated by different combinations of GRNs but also the dynamics of the evoked behavior. Although little is known about the circuits downstream of the GRNs, our analysis of sensorimotor transformations serves as a reference to determine how these circuits organize navigational decision-making.

Although chemotactic navigation behavior involves just two motor states (running and turning), it is possible, in principle, to extend reverse-correlation analysis to a larger number of possible behavioral states. Vogelstein et al presented recently a study where they used optogenetic pulses to trigger different subsets of neurons throughout the larval brain (*Pfeiffer et al., 2008*; *Vogelstein et al., 2014*). They identified 29 statistically different behavioral states, likely because they were able to interrogate circuits for a much wider variety of larval behaviors than navigation. It would be useful to apply reverse-correlation methods such as ours to examine transitions between this rich set of

behavioral states to identify how specific neurons mediate a broader range of behavioral decisions than navigation up or down stimulus gradients.

The wiring diagram of the *Drosophila* larva nervous system is likely to be the next whole animal connectome that will be reconstructed (*Cardona et al., 2010*). Powerful genetic tools are making it possible to target specific neurons throughout the *Drosophila* nervous system with cellular resolution (*Pfeiffer et al., 2008*). The new availability of powerful optogenetic tools for activating and inactivating neurons, particularly red-shifted molecules that are outside the spectrum of *Drosophila* vision, is making it possible to pinpoint the role of specific neurons in overall behavior (*Chuong et al., 2014*; *Klapoetke et al., 2014*). An essential step in building whole nervous system models of behavior that incorporate wiring and dynamics is computational modeling. Bringing together computational modeling of behavior with new tools for behavioral and physiological analysis, such as those described here, should open the door to a thorough understanding of behavioral circuits from sensory input to motor output in the small but surprisingly sophisticated nervous system of the *Drosophila* larva.

## Materials and methods

### *Drosophila* stocks

All larvae were raised in the dark at 22°C and fed yeast with 0.5 mM all-*trans*-retinal. All *GrX-Gal4* lines were previously described (*Weiss et al., 2011*). The *UAS-CsChrimson* flies were a gift of Vivek Jayaraman. Other lines were provided by the Bloomington Stock Center: *Or42a-Gal4* (BL#9970), *Or45a-Gal4* (BL#9975), *Orco-Gal4* (BL#23909), *Gr21a-Gal4* (BL#23890), *Gr66a-Gal4* (BL#28801), and *w1118* (BL#5905).

### Behavioral assays

Male *Gal4* flies were crossed to *UAS-CsChrimson* virgins in small cages (Genesee Scientific, San Diego, CA) where eggs were laid on grape juice plates. Larvae were thoroughly washed in water, and late second-instar larvae were selected under a dissecting microscope. For spatial navigation assays, groups of 20–30 larvae were placed in the center of a ~5-mm thick 22 × 22 cm agar (Fisher Scientific, Pittsburgh, PA) plate and allowed to freely move for 20 min. Animals were recorded with a CCD Mightex camera with a long-pass (740 nm) infrared filter at 4 Hz.

Light stimulation was produced with a custom built LED matrix assembled with SMD 5050 flexible LED strip lights of 12 V DC and 625 nm wavelength (LEDlightninghut.com) and controlled with an H-bridge driver and custom code written for a LabJack U3 controller. Random light sequences were synchronized with the acquisition of images of the camera. Illumination was at 850 nm wavelength with custom built LED bars. The selection of the wavelength of the infrared LEDs was to be far enough from the red LEDs in order to allow the selection of a long-pass filter to avoid the red LED illumination from affecting behavioral recordings.

We mounted the infrared LEDs for dark-field illumination in opto-mechanic elements that allow adjusting the angle of the LED bars with respect to the behavioral arena. This was to avoid larval 'shadows' in the movies, which result in much lower efficiency of data acquisition. The red LEDs were connected in parallel to avoid the creation of a light gradient caused by voltage drop in each LED. We verified uniform light intensity at 1.9 W/m² ± 0.06.

### Electrophysiology

We followed previously described methods (*Kreher et al., 2005*). In brief, action potentials of the ORNs were extracellularly recorded by placing a custom made tungsten recording electrode (with a piezo manipulator) through the cuticle into the dome of the DO of third-instar larvae. The larva was placed on its ventrum on a metal rod and immobilized by wrapping Parafilm around the rod and the body, exposing only the very anterior part of the larva containing the domes of the dorsal organs. A reference electrode, a drawn out borosilicate glass capillary filled with Ephrussi and Beadle solution, was previously inserted through the Parafilm into the larva's body. Light stimulation was generated with an LED at 627 nm (Luxeonstar) driven by a BuckPuck (LUXdrive LEDdynamics) and synchronized via a photocoupler relay (Toshiba TLP597A) with the data acquisition system (Syntech IDAC-4). The electrophysiological optogenetic experiments were

conducted in a completely dark room, and the intensity of the light stimulus at the location of the larva's DO was set to 1.9 W/m².

## Data analysis

The image stacks recorded were processed using the MAGAT (multiple animal gait and trajectory) analyzer (available online at https://github.com/samuellab/MAGATAnalyzer) and analyzed using MATLAB (*Gershow et al., 2012*). To produce the random stimulus, a Bernoulli process was used. This process is wide-sense stationary, produces independent binary values (Lights ON or OFF) at every instant, and its autocorrelation function is the Dirac delta function. The linear transformations for $r \to t$ and $t \to r$ transitions were estimated by the event-triggered averages multiplied by the mean $t \to r$ or $r \to t$ rates, respectively (*Sakai, 1992*; *Dayan and Abbott, 2001*). The convolution of the filters with the stimulus was computed numerically without fitting any function to the filter. The number of larvae used in the experiments of each figure can be found in the respective legends.

## Reverse-correlation calculations

We model navigational behavior as two alternating motor states: runs and turns. This allows the precise quantification of behavioral response as a time series of basic motor patterns. To activate CsChrimson, we use binary ON/OFF red light following a Bernoulli process (see below for rationale for using this process). We assign −1 to OFF state and +1 to ON state. During a run, the behavioral response can be characterized as the likelihood of initiating a turn (thereby ending the run). During a turn, the behavioral response is the likelihood of initiating a new run. Below, we describe our calculations for the run-to-turn transition. The same process was followed to make calculations for the turn-to-run transition. Our calculations follow standard reverse-correlation methods (*Dayan and Abbott, 2001*).

First, we model the animal as a linear transducer. By definition, the probability of transition from run-to-turn, $r_{T \to R}$, is a weighted sum of stimulus history, $s(t)$:

$$r_{R \to T}(t) = \int_0^\infty h_{R \to T}(\tau)s(t - \tau)d\tau, \tag{1}$$

where $h_{R \to T}$ is linear filter or kernel.

The Dirac delta function, $\delta(t)$, is defined by:

$$\int_0^\infty h(\tau)\delta(t - \tau)d\tau = h(t). \tag{2}$$

When the stimulus is a Dirac delta function, the response is a direct measurement of the linear filter:

$$r_{R \to T}(t) = \int_0^\infty h_{R \to T}(\tau)\delta(t - \tau)d\tau = h_{R \to T}(\tau). \tag{3}$$

Alternatively, the first-order filter can be recovered by measuring the response to a random stimulus. The linear filter relates the cross-correlation of input and output ($R_{rs}$) and the autocorrelation of the input ($R_{ss}$) by:

$$R_{rs}(t) = \int_0^\infty h(\tau)R_{ss}(t - \tau)d\tau. \tag{4}$$

For a Bernoulli process, as the one used here, the autocorrelation function ($R_{ss}$) is a Dirac delta function ($\delta(t)$), therefore, *Equation 4* becomes

$$R_{rs}(t) = \int_0^\infty h(\tau)\delta(t - \tau)d\tau = h(t). \tag{5}$$

Thus, the cross-correlation of input and output represents a measurement of the linear filter. In our analysis, the relevant events in the output are the transitions from running to turning and vice versa. The average optogenetic activation signal that precedes each of these events is called the event-triggered average and can be written as shown below:

$$C(\tau) = \frac{1}{\langle n \rangle} \int_0^T r_{R \to T}(t) s(t - \tau) dt, \tag{6}$$

Where $\langle n \rangle$ is the average number of turns per trial, and $T$ is the duration of each trial. In our experiments, trials lasted 20 min.

The cross-correlation function $(R_{rs})$ can be written:

$$R_{rs}(\tau) = \frac{1}{T} \int_0^T r_{R \to T}(t) s(t + \tau). \tag{7}$$

From **Equations (5), (6), (7)**:

$$h(\tau) = R_{rs}(\tau) = \frac{\langle n \rangle}{T} C(-\tau). \tag{8}$$

Thus, we estimated our linear filters by measuring the event-triggered average and multiplying by the average number of events in one trial divided by the duration of the trial.

## Design criteria of white process for random stimulation

While many white processes could potentially be used for reverse-correlation analysis, we selected a Bernoulli process to minimize the introduction of nonlinear relationships between spiking and optogenetic activation. Using Gaussian white noise, for example, would require intensity modulation of the optogenetic stimulus. While it is possible that induced spiking scales linearly with light intensity for some Gal4 drivers, a nonlinear relationship might also occur. By using a Bernoulli process, we can use a uniform light intensity for optogenetic stimulation and need only assume a consistent level of spiking with optogenetic stimulation. This assumption was validated by direct electrophysiological measurement in two ORNs (see **Figure 1**). For a Bernoulli process, the random stimulation is confined to the timing of the ON or OFF state of the LEDs. Nonlinearities might still be introduced, for example, in delays in the onset or cessation of spiking with respect to the start or end of each illumination pulse. These effects pose an upper limit on the frequencies that can be used for random activation. To reduce the effect of these latencies while retaining the ability to characterize larval decision-making that occurs on the 1-s time scale, we used a stimulus frequency of 4 Hz. Because our method accurately predicts behavioral responses to trains of pulses of different widths as the one used in the right panel of **Figure 3C**, the top right panel of **Figure 3—figure supplement 1A**, the right panel of **Figure 3—figure supplement 1B**, or the top right panel of **Figure 3—figure supplement 2A**, nonlinearities owing to spike latencies are not likely to have significantly affected the construction of our models.

## Establishing LN models

To extract LN models and predictions from each 20 min movie of 20–30 animals in each experiment, we followed this workflow:

- Acquire behavioral movies and used the MAGAT Analyzer to segment trajectories.
- Identify all runs and turns and their initial frame and duration.
- Obtain all light patterns preceding the initiation of each run and turn.
- Using the light patterns, compute the triggered average for each transition using **Equation 7** (**Figure 2**).
- Make predictions using the measured triggered average (i.e., convolution of Equation 9 with input signal).
- Compare with experimental measurements and fit sigmoidal function using least squares (e.g., **Figure 3A**).
- Assemble LN model and test using either subjecting a new set of larvae to defined trains of light pulses of different width (e.g., **Figure 3C**, right) or step increases or decreases in illumination (e.g., **Figure 3C**, left).

## Acknowledgements

We thank Vivek Jayaraman for sharing fly stocks, Ni Ji for comments on the manuscript, and Vivek Venkatachalam, Christopher Tabone, Renaud Bastien, Matthew Berck, and Jess Kanwal for useful discussions. This work was supported by grants from the NIH to JC and to ADTS (1P01GM103770 and 8DP1GM105383-05).

# Additional information

## Funding

| Funder | Grant reference | Author |
| --- | --- | --- |
| National Institutes of Health (NIH) | 1P01GM103770 | John R Carlson |
| National Institutes of Health (NIH) | 8DP1GM105383-05 | Aravinthan DT Samuel |

The funder had no role in study design, data collection and interpretation, or the decision to submit the work for publication.

## Author contributions

LH-N, Conception and design, Acquisition of data, Analysis and interpretation of data, Drafting or revising the article, Contributed unpublished essential data or reagents; JB, Conception and design, Acquisition of data, Analysis and interpretation of data; MK, GS, LC, Conception and design, Analysis and interpretation of data, Contributed unpublished essential data or reagents; JRC, Conception and design, Analysis and interpretation of data, Drafting or revising the article; ADTS, Conception and design, Drafting or revising the article

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
