## [Decision Letter]

Thank you for sending your work entitled “Reverse-correlation analysis of sensorimotor dynamics in *Drosophila* larva using optogenetics” for consideration at *eLife*. Your article has been favorably evaluated by Eve Marder (Senior editor) and three reviewers, one of whom, Ronald L Calabrese, is a member of our Board of Reviewing Editors.

The Reviewing editor and the other reviewers discussed their comments before we reached this decision, and the Reviewing editor has assembled the following comments to help you prepare a revised submission.

The authors present a very intriguing behavioral analysis of the navigation of *Drosophila* larvae using optogenetically induced fictive olfactory and gustatory stimuli for specific receptor neurons and for receptor neurons in combination. These stimuli are random on-off (Bernoulli process that is wide-sense stationary and produces independent binary values). In an automated system, they monitor large numbers of larvae to a variety of receptor neuron activations and combinations thereof and apply Linear-Nonlinear (LN) models. Their results predict navigational responses for different optogenetic activation waveforms with reasonable accuracy. They also study the sensorimotor transformations underlying navigation driven by bitter sensing gustatory neurons. They find a diversity in the behavioral dynamics evoked by different subsets of gustatory neurons expressing different taste receptors, suggesting that larvae can discriminate between tastes of the same quality because these evoke kinetically different behavioral responses. These results have important implications for behavioral analysis of navigation in animals and serve as an entry point for further mechanistic studies in the important model system. Moreover, the application of the techniques here can be a model for applications in other systems.

There are concerns about the manuscript as presented that the authors must address, by re-writing and if feasible some additional experiments or more explicit recognition of the limitations of the analysis.

1) The chief concern is with the optogenetic stimuli and their interpretation. The authors say in the Introduction that they are able to “connect sensory activity patterns with motor patterns” via their study. Without recording from the larval chemosensory neurons expressing CsChrimson and determining what patterns of neural activity are evoked by the different optical stimulation protocols, they cannot make this claim. Rather, even though the red light stimulus is a random pattern, the sensory neurons that express CsChrimson may respond in non-linear ways to the red light activation. The patterns of sensory neuron activity therefore may not nicely (linearly) reflect the red light stimulation they provide. It should be feasible to record from the sensory neurons during the optogenetic stimuli and determine how they fire in response. For example using the GR94a (C2) line one could record extracellularly from the dorsal organ and determine how the neuron responds to optical stimuli and in particular to their random flicker pattern of optical stimulation. However, unless a comprehensive e-phys study is performed, the caveats would remain for the untested lines. There must therefore be a section in the Discussion where the authors address the caveats that arise because the extent and quality of the sensory neurons firing to the optogenetic stimulus is unknown. The authors should tone down the claims that turning on the lights is the same thing as turning on the neurons.

Moreover, it is not known how the optogenetic activation equates with neuronal activity in the different genetic backgrounds. A variety of GAL4 drivers are used and the strength of CsChrimson expression among these drivers could be quite different. Thus, one cannot distinguish a weak behavioral response from potentially weak expression of the CsChrimson or to a less important role of the neuron. Thus “the different linear filters” discovered in the study might be simply differences among CsChrimson potency in the cells and this caveat should be stated in the text.

This concern about the relationship between the light stimulus and neuronal activation carries over especially into the interpretation of the 'attractive' response evoked by optogenetic stimulation of the C2 receptor alone. It is quite possible that CsChrimson activation of C2 is very weak and that greater activation would change the valence of the response to aversion. Unless we know the quality of the C2 response to the optogenetic stimuli, it is necessary to be circumspect in the interpretation of C2 mediating an attractive response, and the authors' claim that the hypothesis that “gustatory preferences are… encoded in “valence labeled lines” is weakened by the C2 result is subject to real caveats.

2) One of the authors’ claims is that, essentially, not all attraction or avoidance responses are created equal. That is, different ORNs and GRNs are known to drive “avoidance” or “attraction” but they see that the dynamics of the behavior is different for different neurons (Figure 4). Are there statistical methods that would substantiate these claims of different kinetics?

3) The Materials and methods section is inadequate. It should be substantively amplified to explain more carefully what was done especially with the modeling and the statistical analyses. The reader must assume as the legends of the figures are not explicit (except an off handed mention in Figure 1) that the gray shading around the event triggered average lines in the figure are SEM, but even this is missing or not visible in Figure 5. How do we know for each of these cases, particularly for the case of C2, alone in Figure 5, if these events triggered averages are significant compared to background? What statistical criteria were applied? How were these noisy averages incorporated into the LN model? Please adequately explain what was done. In addition, a more intuitive explanation of the linear-non-linear transformation is needed for readers that are not familiar with this type of modeling.

---

## [Author Response]

*1) The chief concern is with the optogenetic stimuli and their interpretation. The authors say in the Introduction that they are able to* “*connect sensory activity patterns with motor patterns*” *via their study. Without recording from the larval chemosensory neurons expressing CsChrimson and determining what patterns of neural activity are evoked by the different optical stimulation protocols, they cannot make this claim. Rather, even though the red light stimulus is a random pattern, the sensory neurons that express CsChrimson may respond in non-linear ways to the red light activation*. *The patterns of sensory neuron activity therefore may not nicely (linearly) reflect the red light stimulation they provide. It should be feasible to record from the sensory neurons during the optogenetic stimuli and determine how they fire in response. For example using the GR94a (C2) line one could record extracellularly from the dorsal organ and determine how the neuron responds to optical stimuli and in particular to their random flicker pattern of optical stimulation. However, unless a comprehensive e-phys study is performed, the caveats would remain for the untested lines. There must therefore be a section in the Discussion where the authors address the caveats that arise because the extent and quality of the sensory neurons firing to the optogenetic stimulus is unknown*.

We agree. To demonstrate that CsChrimson activates sensory neurons with pulse widths and light intensities comparable to our behavioral measurements, we have now made direct recordings in the dorsal organ as suggested. To do this, we obtained help from Jonas Belina and John Carlson at Yale (who are now coauthors). We found consistent spiking patterns in both Or45a and Or42a over a range of pulse widths comparable to our experiments, confirming the correspondence between optogenetic illumination and sensory neuron activity.

We also agree that spiking triggered by CsChrimson activation, however, might not necessarily mirror optogenetic activation across all tested lines. Thus, we have added discussion of these caveats as recommended.

We note that we were cognizant of the potentially nonlinear relationship between the intensity of optogenetic illumination and spiking. This was why we used a Bernoulli process—i.e., random ON/OFF flicker—as our stimulus. Gaussian white noise is often used in reverse-correlation studies, but would have involved intensity modulation over time, which would have added another nonlinearity in the transformation between optogenetic illumination and behavioral response. By using a Bernoulli process where we fix light intensity, we avoid this problem. The randomness is in the onset or offset of the light, not in various intensity values.

Another consideration we made when selecting the random process is the possibility that there could be delays in onset of offset of spiking with respect to CsChrimson activation. This constrains the frequencies that can be used for optogenetic stimulation; since the timescale for behavior is ∼1 second, we used a frequency of 4Hz. Furthermore, if significant time delays in the onset or offset of spiking occur with respect to CsChrimson activation, then the triggered average of optogenetic activation would likely fail to predict the behavioral responses to trains of pulses of different widths as the one used in the right panel of Figure 3, the top right panel of Figure 3—figure supplement 1, the right panel of Figure 3—figure supplement 1, or the top right panel of Figure 3—figure supplement 2.

*Moreover, it is not known how the optogenetic activation equates with neuronal activity in the different genetic backgrounds. A variety of GAL4 drivers are used and the strength of CsChrimson expression among these drivers could be quite different. Thus, one cannot distinguish a weak behavioral response from potentially weak expression of the CsChrimson or to a less important role of the neuron. Thus* “*the different linear filters*” *discovered in the study might be simply differences among CsChrimson potency in the cells and this caveat should be stated in the text*.

We agree. We now comment on this caveat in a paragraph of the Discussion.

*This concern about the relationship between the light stimulus and neuronal activation carries over especially into the interpretation of the 'attractive' response evoked by optogenetic stimulation of the C2 receptor alone. It is quite possible that CsChrimson activation of C2 is very weak and that greater activation would change the valence of the response to aversion. Unless we know the quality of the C2 response to the optogenetic stimuli, it is necessary to be circumspect in the interpretation of C2 mediating an attractive response, and the authors' claim that the hypothesis that* “*gustatory preferences are… encoded in* “*valence labeled lines*” *is weakened by the C2 result is subject to real caveats*.

We agree. We have removed the discussion of “valence labeled lines”. Confirmation of the role of these neurons might be possible by identifying a chemical stimulus specific to Gr94a, and then analyzing chemotaxis in response to that stimulus. Dissecting these interesting issues in neurobiology are beyond the scope of our article, which is to present a new method for triggering and analyzing behavioral dynamics.

*2) One of the authors’ claims is that, essentially, not all attraction or avoidance responses are created equal. That is, different ORNs and GRNs are known to drive* “*avoidance*” *or* “*attraction*” *but they see that the dynamics of the behavior is different for different neurons (*Figure 4*)*. *Are there statistical methods that would substantiate these claims of different kinetics?*

This is a great suggestion. We added a statistical analysis to Figure 4 (also see Figure 4—figure supplement 1) that clearly demonstrates the dynamic differences between Or45a and Gr21a, and also between Or42a and Gr2a. We have added text that explains the differences in behavioral dynamics in terms of our statistical analysis.

*3) The Materials and methods section is inadequate. It should be substantively amplified to explain more carefully what was done especially with the modeling and the statistical analyses. The reader must assume as the legends of the figures are not explicit (except an off handed mention in*
Figure 1*) that the gray shading around the event triggered average lines in the figure are SEM, but even this is missing or not visible in*
Figure 5.

Thank you for the suggestion. We have expanded the Methods to explain in detail the calculations used for the LN model construction, the technical recommendations for building an experimental setup as ours, and the considerations for the analysis of behavior made in this study. We have also improved Figure 5 to show the grey shaded regions (although they remain small because of the large amount of data collected). We have augmented the figure legends as recommended.

*How do we know for each of these cases, particularly for the case of C2, alone in*
Figure 5*, if these events triggered averages are significant compared to background? What statistical criteria were applied? How were these noisy averages incorporated into the LN model? Please adequately explain what was done. In addition, a more intuitive explanation of the linear-non-linear transformation is needed for readers that are not familiar with this type of modeling*.

We have run additional statistical analysis to show that the case of C2 alone and C1 alone are significantly different than the control. These results are shown in a new figure (Figure 5—figure supplement 1).

In the new expanded Methods we have included a detailed explanation about how the integration is conducted and a simple introduction to the type of modeling we used.